## [Peer Review File · Nature Structural & Molecular Biology]

UBA6 specificity for ubiquitin E2 conjugating enzymes reveals a priority mechanism of BIRC6

Corresponding Author: Dr Paul Elliott

Version 0:

Decision Letter:

2nd May 2025

Dear Dr. Elliott,

Thank you again for submitting your manuscript "Molecular basis of UBA6 specificity for ubiquitin E2 conjugating enzymes reveals a priority mechanism of BIRC6". We now have comments (below) from the 3 reviewers who evaluated your paper. In light of these reports, we remain interested in your study and would like to see your response to the comments of the referees, in the form of a revised manuscript.

You will see that though all reviewers appreciate the potential value of the manuscript, they nevertheless raise notable concerns and provide very relevant feedback that needs to be comprehensively heeded in a revised manuscript. All three experts note the absence of cellular data validating the biological significance of the findings and they also bring up the importance of obtaining such results to showcase the validity and physiological relevance of the proposed mechanisms. We agree with the reviewers that such experiments, with the recommendations for co-IPs and relevant functional data being pertinent guidance, will boost the value of the manuscript and therefore request that you perform them. Furthermore, Reviewer #3 and Reviewer #1 propose biochemical experiments (introducing mutations in the blocking loop in BIRC6 and SEC analyses, respectively) to further support the proposed mechanistic models. Finally, all experts note the importance of adding relevant clarifications, further citations and missing information, and further discussing by answering relevant conceptual questions.

Please be sure to address/respond to all concerns of the referees in full in a point-by-point response and highlight all changes in the revised manuscript text file. If you have comments that are intended for editors only, please include those in a separate cover letter.

We expect to see your revised manuscript within 3-4 months. If you cannot send it within this time, please contact us to discuss an extension; we would still consider your revision, provided that no similar work has been accepted for publication at NSMB or published elsewhere.

Reporting Summary:

EXTENDED DATA FIGURES

Data availability: this journal strongly supports public availability of data. All data used in accepted papers should be available via a public data repository, or alternatively, as Supplementary Information. If data can only be shared on request, please explain why in your Data Availability Statement, and also in the correspondence with your editor. Please note that for some data types, deposition in a public repository is mandatory - more information on our data deposition policies and available repositories can be found below:

<https://www.nature.com/nature-research/editorial-policies/reporting-standards#availability-of-data>

Link Redacted

Sincerely,

Dimitris Typas
Senior Editor
Nature Structural & Molecular Biology
ORCID: 0000-0002-8737-1319

Reviewers' Comments:

Reviewer #1 (Remarks to the Author):

The manuscript by Riechmann et al. is a rich study exploring the UBA6-dependent ubiquitylation system, which is orthogonal to the canonical UBA1-dependent pathway in vertebrates. The study is well-designed with a combination of cryoEM and clever biochemistry, and is rigorously executed with appropriate assays and controls, and - to the extent that I can judge it - appropriate statistics. References to previous work seem exhaustive enough. The cryoEM data (lots of it!) seem reliably processed, but I am mostly a crystallographer myself so might have overlooked something. The proposed mechanistic insights – including the final model for how BIRC6 outcompetes other E2s through high-affinity binding but is, at the same time, prevented from overstaying on UBA6 through an apparent dedicated release mechanism – seem well supported (perhaps it could just be stressed somewhere that these insights/models are based on in vitro biochemistry and how they play out in cells would additionally depend on protein levels, colocalisation or lack thereof in the same cells and cell areas, etc.). The length of the manuscript seems appropriate – if anything, the manuscript might even seem too compact, but the authors have made effort not to overload it, confining some details (e.g. some amino-acid residue and mutation details) to figures rather than the main text, as well as using schematics and including some parts as extended data – I think all this was skilfully done. Overall, my impression is positive, and I think this interesting and rigorous study should be published. While appreciating the study requires knowing some intricacies of the ubiquitylation pathway, ubiquitylation is a very large field, so the study should be of fairly broad interest, and it would interest people from some other disciplines, at least those studying other ubiquitin-family modifications, which are mechanistically similar.

Minor points

- p.2, line 30: I think “this” is doing a bit too much in this sentence. This part of the abstract would benefit from saying more clearly that high-affinity binding, while allowing BIRC6 to gain priority over other E2s, could lead to the stalling of BIRC6 on UBA6 thus potentially inhibiting the enzyme, and it is this aspect – and not the preference over other E2s as such – that is apparently countered by a release mechanism whereby BIRC6, once ubiquitylated, binds more weakly to UBA6. In short, I think the abstract could be slightly clearer. The first time I read it, I didn't understand everything.
- p. 3, line 93: the authors could consider commenting on the apparent role of divergent and convergent evolution in compatibility of E2s with E1s
- p. 5, lines 195-6: I feel like there is a problem in this sentence with plural being mixed with singular (“... that regulate their activity and ... interacts with”)
- p. 5, line 211: “evolutionary-guided” – would be “evolution-guided” sound better? The authors will be able to judge this better than I though
- p. 6, line 236: “Consistent with all other ubiquitin E2s” – do the authors mean all others except for human BIRC6? This could be clarified.
- p. 7, line 331: “This suggests that BIRC6UBC in fact binds UBA6 stronger when the Cys-Cap loop is removed” – would one see this stronger binding in analytical SEC like that in Fig. 2a, perhaps as a more complete leftward shift of the BIRC6UBC band? Or alternatively one could consider another technique for comparing the binding to WT and Cys-Cap-deleted UBA6. I don't think this is a must but would strengthen the final part of the study. I feel like some direct insight into binding would add extra value to the clever competition assays.
- p. 8, line 358: “unique” among which processes? Perhaps “unique among PTMs”?
- p. 8, lines 360: the authors could try avoiding saying “UBA6-activated/specific ubiquitylation” twice in one sentence
- p. 8, line 382 – maybe the authors could define what they mean by extension – extension with respect to what reference UBC domain and how long would it have to be to count as an extension? Should one distinguish between E2s having extra structured domains and just having extra short flexible tails etc.?
- a thought for the discussion: can there be any other reasons why BIRC6 has developed high affinity for UBA6, other than competition with other E2s? Moreover, the authors speak about a stable complex, as judged by SEC, so that might suggest not just fast association but also slow dissociation - can one speculate about some special reasons why would that be? I was intrigued by the cited observation that only a part of UBA6 is Ub-loaded at any given time in the cell. Could UBA6 in its non-loaded state bind to BIRC6 and the two wait together for loading and transthiolation – in this case, would the BIRC6-bound state of UBA6 be equally, or more, or less efficient at binding Ub and ATP and activating Ub? Lastly, how is all this timed with substrate modification - what is the kinetic bottleneck in this cascade? - and could substrate binding further help detach Ub~BIRC6 from UBA6?
- this is probably silly and not worth discussing but could pharmacological strategies be envisioned to trap BIRC6 on UBA6 to target UBA6-dependent cancer? I feel like wherever in biology we have a system that must be released to continue, there is this theoretically potential for getting it trapped, like with PARP1.
- p. 51, line 1117: the authors mention that the noncovalent complex actually comes from a UbDha-trapping approach but with a probe with reduced activity. It is good of authors to provide this detail (perhaps it could be explained slightly more), and I wonder if this should be signalled briefly in the main text and figure legend for complete transparency.

Further suggestions

I don't think it is a must, and I'm sure the authors have considered it already, but their model would benefit from having a stable mimetic of ubiquitylated BIRC6 or BIRC6 fragment. This would allow directly investigating the binding of such a mimetic – provided it mimics well – to UBA6 in experiments similar to that in Fig. 2a, as well as investigating the competition between ubiquitylated BIRC6 and other E2s in experiments similar to those in Fig. 2g and Fig. 5.

Secondly, I wonder if it might be possible to investigate the kinetic parameters (k_{on} , k_{off}) of BIRC6 – and potentially also ubiquitylated BIRC6 – binding to UBA6.

In cells, some validation of the model could potentially be gained through co-immunoprecipitation experiments, demonstrating competition between E2s or lack thereof depending on the WT/mutant variants of UBA6 and BIRC6 used. A more final proof would surely involve doing some functional cellular assay to see, for example, if BIRC6 that has lower affinity for UBA6 is not functional.

Having listed these suggestions, I don't think these experiments are required; these are just suggestions that came to my mind for how the model could be further tested and expanded, although the authors have thought much longer about it than me and I'm sure they have better ideas for further validating and developing this work.

Reviewer #2 (Remarks to the Author):

Review for NSMB:

This is a very interesting and comprehensive biochemical and structural study investigating how UBA6, which is a second E1 for ubiquitin, and much less well understood than UBA1, functions with E2s. Their careful analyses lead them to propose a hierarchy of E2 preference by E1s, which in my opinion fits well with and extends early work in the field which established how E1's ubiquitin-like fold functions with specific ubls and E2s. The finding that different E2s need different contributions of E1 architecture is useful.

The data presented are high quality and convincing. The structures are relatively limited in resolution with heterogeneity in ubiquitin conformations in the quaternary complex structures, but the aspects focussed on in the biochemical validation are reasonable and justified.

They show that 2 E2s (2Z and BIRC6) are specific for UBA6, and use different mechanisms to achieve the specificity.

In my opinion this study answers an important question as to how the two E1s for ubiquitin work with multiple E2s, from a molecular perspective. As well as this general insight, the main novel finding is the thioester switch mechanism for BIRC6~Ub release – this is an important contribution to understanding E1-E2 interactions and unique adaptations of UBA6 to function with BIRC6.

I think it could be argued that the evolutionary analysis is somewhat limited, focussing on Drosophila and human – what about other invertebrates? There is also no functional assay in a biological context to explore how the hierarchy they propose works. However, I would consider that outside the scope of this study, which is already a significant contribution.

Minor points

I'm not sure 'refined to' is the correct term in cryoEM, it is for crystallography, but there is no corresponding 'refinement' in cryoEM.

First paper to report ubiquitin fold domain in an E1, and to show that it was required for E2 binding is PMID: 12646924 from 2003, from the Schulman lab.

Some of the quoted figures seem a bit too precise for the resolution – 899 angstrom squared buried surface area or 214 A² at lower than 3A resolution might be better expressed to the nearest 10 angstroms than the nearest one.

Line 169 – abrogated is a legal term, can the authors just say it reduced or prevented binding? The writing is a little dense in parts, maybe because the text was single spaced it gives me that impression, but I had to read the structure sections several times to get the information.

Discussion – I don't really understand what a molecular 'rulebook' is – I think they are trying to convey that they have worked out the molecular basis of the specificity of each E1 for its E2s, and that knowing that allows for protein engineering, perhaps framework is a less ambiguous term (and allows for possible future findings that may add to the current understanding!).

Papers associated with pdb files should be cited (eg in figure legends) as they are experimentally determined and reported.

Extended figure 1 legend doesn't include UBA7

Why do the PDB validation reports say 'not for manuscript review'? Am I supposed to use them in the review or not?

Reviewer #3 (Remarks to the Author):

In the present manuscript, Riechmann et al investigate specificity determinants governing ubiquitin E1 and E2 enzyme interplay. Through elegant bioinformatic, molecular, and biochemical approaches the authors systematically define sets of E2 enzymes that can be specifically charged by either of the two ubiquitin activating enzymes, UBA1 or UBA6, and identify the E1 domains underlying this specificity. Consistent with previous results, they find UBE2Z and BIRC6 to be specifically charged by UBA6. They then focus on characterizing the interaction between UBA6 and BIRC6, an E1-E2 pair with previously identified strong genetic interactions. By performing biochemical, biophysical, and cyro-EM analyses, they demonstrate that UBA6-BIRC6 form an unusually strong complex in the absence of ATP and ubiquitin (as compared to other E1 and E2s) and that complex formation is driven by the BIRC6UBC-UBA6UFD domain interaction. This binding mode provides BIRC6 with priority for UBA6 over recognition of other E2s, as demonstrated by in vitro competition experiments. To

delve deeper into understanding specificities underlying the UBA6-BIRC6 interaction, the authors next employ an evolutionary-guided approach. They identify that the *Drosophila melanogaster* genome encodes a BIRC6 orthologue (dBIRC6) but only one E1 orthologue that is more similar to UBA1 (dUBA1). They can show that dBIRC6 is charged by dUBA1 and, intriguingly, human UBA1 is able to charge dBIRC6. They then leverage this cross-species E1 reactivity and determine and cryoEM structure of human and drosophila E1-UbDha-BIRC6UBC complexes that capture transthiolation intermediates and help explain how UBA6-BIRC6 specificity is achieved. Finally, through analysis of their cryoEM structures coupled with elegant mutational validation experiments, the authors provide evidence for a thioester-switch mechanism as a molecular rationale for how BIRC6, despite its high affinity to UBA6, can be released after ubiquitin transfer.

This is a well-executed and important study, which identifies a previously unrecognized priority mechanism for UBA6-mediated ubiquitin transfer to BIRC6, providing a biochemical rationale for the known strong UBA6-BIRC6 genetic interaction. In addition, this study reveals novel molecular and structural details underlying activity of subset of ubiquitin E2s with their cognate E1s, UBA1 and UBA6, offering much needed insights into mechanisms that may allow for E1s to orchestrate E2-specific cellular functions. Given the emerging role of E1 enzyme mutations in several human diseases (i.e., spinal muscular atrophy, lung cancer in never smokers, and VEXAS), the results of this study will be of broad interest not only to researchers in the ubiquitin field, but also to those studying protein homeostasis, cell signaling, and disease mechanisms more generally.

Below, I am providing questions and suggestion to improve this already very strong study:

1) Molecular rulebook for E1-E2 specificity:

- carefully examining extended data figure 2, I am wondering about the criteria that were used for classifying UBA1-specific E2s in UFD- and UFD- /SCCH-dependent categories, respectively. I.e., were there specific cutoffs used for residual activity with E1 chimera? By eye, it looks like UBE2C still has activity with chimera ii and iv, which is consistent with it being UFD-dependent only. Could the authors please clarify?
- I understand that for dual specific E2s subclassifying into UFD- and UFD/SCCH-dependent categories is not easily feasible. However, looking over the charging pattern of these E2s with the different chimeras, it looks like that for some E2s (e.g., UBE2L3), UFD and SCCH requirements differ depending on UBA1 and UBA6 context, suggesting that for the same E2, UBA6 and UBA1 may have evolved different recognition strategies. This may be worth discussing in the text.

2) Full length BIRC6:

- As mentioned by the authors, BIRC6 is a ~530kDa protein with many N-terminal domains before the UBC domain. Apart from the initial ubiquitin charging experiments determining that BIRC6 is a UBA6-specific E2 enzyme relying on both UFD and SCCH domain of UBA6 (Extended data figure 2c), all experiments have been performed with the BIRC6UBC domain only. While this is reasonable given the challenges working with such a big protein biochemically, I think it would greatly improve the manuscript if at least some of the competition experiments that address the priority ubiquitin charging mechanism would be repeated with full length BIRC6.
- since a full-length BIRC6 structures (albeit with limited UFD resolution) are available, could the authors conduct molecular modeling/docking studies or at least discuss whether their proposed mechanism for ubiquitin charging of BIRC6 by UBA6 is compatible with the full length BIRC6? In addition, would such studies indicate potential contributions of N-terminal domains of BIRC6 to UBA6-specific charging?

3) Priority mechanism and thioester switch mechanism:

- While the data presented for the UBA6-BIRC6 priority mechanism is convincing in vitro, some additional evidence from cells would further strengthen this study. E.g. are there previously published immunoprecipitations experiments showing that BIRC6 is the most abundant E2 enzyme interacting with UBA6 (despite competing with all other UBA6-dependent E2s), or could the authors perform such experiments? Along these lines, compared to WT, would IP of the Cys-Cap UBA6 mutant result in stronger BIRC6 interaction from cells?
- Could the authors further strengthen their thioester-switch mechanism by introducing mutations in the blocking loop in BIRC6 and providing biochemical evidence for its proposed role in reducing the affinity of the BIRC6~Ub-UBA6 interaction?
- As mentioned by the authors, a recent study (PMID: 39143218) has shown that E1-to-E2 transthiolation is driven by coupling adenylation and transthiolation. Could the authors discuss in a bit more detail how this would integrate with their model of the thioester switch mechanism for UBA6?
- Out of curiosity, since the E1-E2 and E2-E3 surfaces partially overlap, would the various E2 mutants that modulate E1 specificity generated by this study also interfere with E3 interactions?

Statistical analysis:

- Statistical analysis should be included for Fig 1c, Fig 3c,e,f, Fig 5c,e

Figure reference and legend errors and suggestions to improve figures:

- Fig 3a: please explain the color coding of the residues in the legend.
- Fig 3b: the words "additional FITC-labelled gel image" in the figure legend should go with 3c
- Fig 3b: consider highlighting the steric clashes and favorable interactions with some dotted lines or distance information
- In section "BIRC6-UBC-UBA6-UFD interaction explain specificity for UBA1-specific E2s" extended figure 3c should refer to 2c
- Extended data figure 1 legend indicates that UBA7 is in the alignment, but UBA7 is not shown. There are some rows with UBA7 labels. Please correct this figure and also explain color coding
- Extended data figure 8: the figure legends do not match up with the legends. Please also indicate which experiments are

single or multi-turnover reaction. Should be cleared up.

Version 1:

Decision Letter:

Our ref: NSMB-A50827A

7th Oct 2025

Dear Dr. Elliott,

Thank you for submitting your revised manuscript "Molecular basis of UBA6 specificity for ubiquitin E2 conjugating enzymes reveals a priority mechanism of BIRC6" (NSMB-A50827A). It has now been seen by the original referees and their comments are below. The reviewers find that the paper has improved in revision, and therefore we can now accept it in principle in Nature Structural & Molecular Biology, pending revisions to satisfy the referees' final requests and to comply with our editorial and formatting guidelines.

To facilitate our work at this stage, it is important that we have a copy of the main text without the figures as a word file. If you could please send along a word version of this file as soon as possible, we would greatly appreciate it; please make sure to copy the NSMB account (cc'ed above).

Sincerely,

Dimitris Typas
Senior Editor
Nature Structural & Molecular Biology
ORCID: 0000-0002-8737-1319

Reviewer #1 (Remarks to the Author):

I thank the authors for their thoughtful responses and for revising their manuscript. The new experiments, including those involving the thioester mimetic and the blocking loop mutants - and the two new approaches (fluorescence anisotropy and cellular pulldown) - complement and strengthen the manuscript. Below, I list a few last minor things I noticed; I think the manuscript is otherwise ready for acceptance.

verse 147: "which" missing before "could"?

verse 203: "analyses offers" -> "analyses offer"

verse 338: Could the authors illustrate how the mimetic was made using UbDha (a similar scheme to that in Ext Dat Fig 11d)? Perhaps it is provided somewhere but I missed it.

verse 339: The 1.5-fold difference in binding between the Ub-BIRC6 mimetic and BIRC6 alone is modest, especially given the relatively large errors in the estimated K_d values provided in Fig. 5c. I think it is still consistent with the authors' model, but they could comment on the remaining uncertainty. It might be worth giving the estimated K_d values together with errors in the main text and not only in the figure.

verse 403: Consider adding the part between asterisks for better clarity: "However, this increase in binding is still not sufficient to enable priority *of UBE2Z or BIRC6 A4575D* over UBE2D2 (Fig. 6e,f) compared to BIRC6 WT".

verse 518: The sentence starting with the following phrase is a bit convoluted - consider revising and perhaps dividing into two sentences: "We therefore propose that the Cys-Cap loop may...".

verse 966: This is not important, but it might be interesting to do ConSurf colouring based on MSA of BIRC6 orthologues only from species that have UBA6.

verse 1281: "no K_d could be fit" -> "no K_d could be fitted"

verse 1508: there is a rectangle symbol after MgCl (instead of 2 in subscript?)

Suggestion for the future

I believe this goes beyond the scope of the current manuscript, but I wonder if the fluorescence anisotropy assay could be used in the future to test displacement of fluorescently-labelled BIRC6 or Ub-BIRC6 by their unlabelled counterparts provided in excess to estimate K_{off} values for the labelled compounds. One could also test displacement of Ub-BIRC6 by BIRC6. Furthermore, I wonder if it might be worth repeating the measurement of K_ds for Ub-BIRC6:UBA6 and BIRC6:UBA6 in the presence of an excess of free Ub that might help outcompete Ub-BIRC6 in a physiologically-relevant manner.

Reviewer #2 (Remarks to the Author):

The authors have satisfactorily addressed all my previous comments and concerns.

Reviewer #3 (Remarks to the Author):

In the revised version of their manuscript, Riechmann et al. have done an excellent job in carefully and comprehensively addressing all of the points I raised in my original review. In particular, they clarified classification criteria and expanded on mechanistic distinctions for dual-specific E2s, repeated key competition assays with full-length BIRC6, and strengthened their proposed priority and thioester-switch mechanisms with new biochemical and cellular evidence. This is an important study that not only provides new mechanistic insight but also generates valuable tools to study E1–E2 interactions in the future. I congratulate the authors on their impactful work.

Version 2:

Decision Letter:

22nd Oct 2025

Dear Dr. Elliott,

We are now happy to accept your revised paper "UBA6 specificity for ubiquitin E2 conjugating enzymes reveals a priority mechanism of BIRC6" for publication as an Article in Nature Structural & Molecular Biology.

Your paper will be published online soon after we receive proof corrections and will appear in print in the next available issue. You can find out your date of online publication by contacting the production team shortly after sending your proof corrections.

If you have not already done so, we strongly recommend that you upload the step-by-step protocols used in this manuscript to the Protocol Exchange. Protocol Exchange is an open online resource that allows researchers to share their detailed experimental know-how. All uploaded protocols are made freely available, assigned DOIs for ease of citation and fully

searchable through nature.com. Protocols can be linked to any publications in which they are used and will be linked to from your article. You can also establish a dedicated page to collect all your lab Protocols. By uploading your Protocols to Protocol Exchange, you are enabling researchers to more readily reproduce or adapt the methodology you use, as well as increasing the visibility of your protocols and papers. Upload your Protocols at www.nature.com/protocolexchange/. Further information can be found at www.nature.com/protocolexchange/about.

Authors may need to take specific actions to achieve compliance with funder and institutional open access

mandates. If your research is supported by a funder that requires immediate open access (e.g. according to <https://www.springernature.com/gp/open-science/plan-s-compliance> Plan S principles or the <https://www.springernature.com/gp/open-science/us-federal-agency-compliance> NIH public access policy) then you should select the gold OA route, and we will direct you to the compliant route where possible. Because authors warrant under our subscription licensing terms that they haven't committed to licensing any version of their article under a licence inconsistent with the terms of our agreement – including the applicable embargo period – publication under the subscription model isn't suitable for authors whose funders require no embargo.

Sincerely,

Dimitris Typas
Senior Editor
Nature Structural & Molecular Biology
ORCID: 0000-0002-8737-1319

Open Access This Peer Review File is licensed under a Creative Commons Attribution 4.0 International License, which permits use, sharing, adaptation, distribution and reproduction in any medium or format, as long as you give appropriate credit to the original author(s) and the source, provide a link to the Creative Commons license, and indicate if changes were

made.

Thank you again for submitting your manuscript "Molecular basis of UBA6 specificity for ubiquitin E2 conjugating enzymes reveals a priority mechanism of BIRC6". We now have comments (below) from the 3 reviewers who evaluated your paper. In light of these reports, we remain interested in your study and would like to see your response to the comments of the referees, in the form of a revised manuscript.

You will see that though all reviewers appreciate the potential value of the manuscript, they nevertheless raise notable concerns and provide very relevant feedback that needs to be comprehensively heeded in a revised manuscript. All three experts note the absence of cellular data validating the biological significance of the findings and they also bring up the importance of obtaining such results to showcase the validity and physiological relevance of the proposed mechanisms. We agree with the reviewers that such experiments, with the recommendations for co-IPs and relevant functional data being pertinent guidance, will boost the value of the manuscript and therefore request that you perform them.

Furthermore, Reviewer #3 and Reviewer #1 propose biochemical experiments (introducing mutations in the blocking loop in BIRC6 and SEC analyses, respectively) to further support the proposed mechanistic models. Finally, all experts note the importance of adding relevant clarifications, further citations and missing information, and further discussing by answering relevant conceptual questions.

We thank the reviewers for their time and constructive critique of our manuscript. We are delighted that they share our enthusiasm for our mechanistic insights into E1-E2 specificity and the unique interaction between UBA6 and BIRC6. We found all comments very insightful and have incorporated all suggestions to strengthen our manuscript. We outline our additional findings below before responding to each comment.

Measuring affinity of the full-length UBA6–BIRC6 interaction

We developed a fluorescence polarisation binding assay using fluorescently labelled E2s and used this to interrogate the mechanistic basis of BIRC6 priority:

1. Building upon our ITC data detecting a tight affinity of BIRC6 to the isolated UBA6 UFD domain, we detected a strong nanomolar interaction between BIRC6 UBC and full-length UBA6 (**Fig. 2g**). There was no detectable interaction with the only other UBA6-specific E2, UBE2Z, or the UFD null-binding mutant introduced into BIRC6 (A4575D).
2. We generated a stable mimetic of BIRC6 post-transthiolation through our UbDha loading strategy and determined affinities to UBA6 variants. BIRC6–Ub bound weaker to UBA6 compared to BIRC6 alone supporting the thioester switch mechanism for releasing BIRC6~Ub thus preventing BIRC6 inhibition of UBA6.
3. Deleting the Cys-Cap loop from UBA6 increased the affinity to BIRC6 nearly 20-fold (**Fig. 6d**). Furthermore, BIRC6~Ub bound UBA6 Δ Cys-Cap loop tighter than UBA6 WT. Together, this explains how BIRC6 priority is increased further upon loss of the Cys-Cap loop and highlights the requirement of the Cys-Cap loop to dampen E2 affinities, which is imperative in the case of tight-binding BIRC6 to avoid inhibition of UBA6 (**Fig. 6g**).

Mutations in the BIRC6 blocking loop to support the thioester switch mechanism

We identified catalytically inactive BIRC6 (C4666A) out-competes UBE2D2 better than BIRC6 WT, supporting action of a thioester switch in BIRC6~Ub release. This data is now displayed more prominently in **Fig. 5 a,b**. We observed the BIRC6 blocking loop undergoes a conformational change upon ubiquitin transfer to BIRC6 active site. Within this loop, a highly conserved Trp (Trp4673) occludes the active site cysteine and must reposition for transthiolation to occur. We made three constructs altering the BIRC6 blocking loop and assessed their ability to outcompete UBE2D2 and bind UBA6. Through these mutations we have further delineated the contribution of the BIRC6 blocking loop:

1. Trp 4673 to Ala (which we predicted to negate the need for dramatic conformational change of the blocking loop) showed a slight enhanced competition compared to BIRC6 WT or C4666A. This is consistent with a slightly increased affinity to UBA6.
2. Trp 4673 to Arg (which we predicted to cause a greater conformational change in the blocking loop and surrounding region) showed weaker out-competition of UBE2D2 loading than WT or C4666A. However, BIRC6 W4673R displayed higher out-competition than BIRC6 A4575D (UFD null-binding mutant) even though no binding was detected for either mutant.

3. Deletion of the blocking loop, including Trp4673, resulted in a slight increased competition of UBE2D2 loading relative to BIRC6 WT and C4666A even though the affinity between BIRC6 Δ blocking loop and UBA6 was reduced \sim 2 fold. This supports a role of the blocking loop in mediating BIRC6~Ub release.

Confirming UBA6-BIRC6 interaction in cell-based assays

By transfecting StrepII-GFP-tagged UBA6 variants into HEK293T cells we detected endogenous BIRC6 with appropriate UBA6 variants supporting our biochemical, biophysical and structural data:

1. Endogenous BIRC6 was detected upon pull-down of wild-type full-length UBA6 which increased with catalytically inactive UBA6 demonstrating BIRC6 is promptly released after transthiolation; thus the thioester switch mechanism occurs in cells
2. BIRC6 was significantly enriched with UBA6 lacking the Cys-Cap loop (either catalytically active or inactive) recapitulating our in vitro biochemical data identifying an increased binding affinity both pre- and post-transthiolation (**Fig. 6d**).
3. BIRC6 was not detected with UBA6-1UFD in which the UFD domain is swapped from UBA6 to UBA1 preventing BIRC6 binding (**Extended Figure 2c**).

Below we respond to individual reviewer comments.

Reviewer #1 (Remarks to the Author):

The manuscript by Riechmann et al. is a rich study exploring the UBA6-dependent ubiquitylation system, which is orthogonal to the canonical UBA1-dependent pathway in vertebrates. The study is well-designed with a combination of cryoEM and clever biochemistry, and is rigorously executed with appropriate assays and controls, and - to the extent that I can judge it - appropriate statistics.

Once again, we thank the reviewer for taking the time to critique our manuscript and appreciate their positive feedback.

References to previous work seem exhaustive enough. The cryoEM data (lots of it!) seem reliably processed, but I am mostly a crystallographer myself so might have overlooked something. The proposed mechanistic insights – including the final model for how BIRC6 outcompetes other E2s through high-affinity binding but is, at the same time, prevented from overstaying on UBA6 through an apparent dedicated release mechanism – seem well supported (perhaps it could just be stressed somewhere that these insights/models are based on in vitro biochemistry and how they play out in cells would additionally depend on protein levels, colocalisation or lack thereof in the same cells and cell areas, etc.).

We agree with the reviewer that, in a cellular context, factors like protein levels, sub-cellular localisation, and post-translational modifications can all affect interactions. Our inclusion of expressing UBA6 variants in cells complements our biochemical evidence showing that 1) UBA6 lacking the Cys-Cap loop binds to BIRC6 more tightly 2) a thioester switch release mechanism exists, as indicated by increased pull-down of BIRC6 with UBA6 C625A (**Fig. 6g**).

The length of the manuscript seems appropriate – if anything, the manuscript might even seem too compact, but the authors have made effort not to overload it, confining some details (e.g. some amino-acid residue and mutation details) to figures rather than the main text, as well as using schematics and including some parts as extended data – I think all this was skilfully done. Overall, my impression is positive, and I think this interesting and rigorous study should be published. While appreciating the study requires knowing some intricacies of the ubiquitylation pathway, ubiquitylation is a very large field, so the study should be of fairly broad interest, and it would interest people from some other disciplines, at least those studying other ubiquitin-family modifications, which are mechanistically similar.

We thank the reviewer for their positive feedback.

Minor points

- p.2, line 30: I think “this” is doing a bit too much in this sentence. This part of the abstract would

benefit from saying more clearly that high-affinity binding, while allowing BIRC6 to gain priority over other E2s, could lead to the stalling of BIRC6 on UBA6 thus potentially inhibiting the enzyme, and it is this aspect – and not the preference over other E2s as such – that is apparently countered by a release mechanism whereby BIRC6, once ubiquitylated, binds more weakly to UBA6. In short, I think the abstract could be slightly clearer. The first time I read it, I didn't understand everything.

We thank the reviewer for mentioning this. We have adjusted the text accordingly to avoid ambiguity.

- p. 3, line 93: the authors could consider commenting on the apparent role of divergent and convergent evolution in compatibility of E2s with E1s

We have adjusted the text to reflect how within the E2J and E2G families of E2s have evolved to function with UBA6.

- p. 5, lines 195-6: I feel like there is a problem in this sentence with plural being mixed with singular (“... that regulate their activity and ... interacts with”)

Thank you, this has been corrected.

- p. 5, line 211: “evolutionary-guided” – would be “evolution-guided” sound better? The authors will be able to judge this better than I though

Thank you, this has been corrected.

- p. 6, line 236: “Consistent with all other ubiquitin E2s” – do the authors mean all others except for human BIRC6? This could be clarified.

Thank you, this has been corrected.

- p. 7, line 331: “This suggests that BIRC6UBC in fact binds UBA6 stronger when the Cys-Cap loop is removed” – would one see this stronger binding in analytical SEC like that in Fig. 2a, perhaps as a more complete leftward shift of the BIRC6UBC band? Or alternatively one could consider another technique for comparing the binding to WT and Cys-Cap-deleted UBA6. I don't think this is a must but would strengthen the final part of the study. I feel like some direct insight into binding would add extra value to the clever competition assays.

We thank the reviewer for an excellent suggestion, and accordingly, we decided to implement a fluorescence polarisation (FP) assay. We chose this method over SEC after observing that BIRC6 binds UBA6 strongly enough to co-elute in analytical SEC. Therefore, subtle changes in affinity might not be detectable. The FP assay reveals a nanomolar interaction between BIRC6 and UBA6, not present with UBE2Z (**Fig. 2g**), and we extrapolate these findings to other E2s given our observed BIRC6 priority over all other E2s (**Fig. 2h and Extended Figure 8c**). Interestingly, deletion of the Cys-Cap loop enhances BIRC6 binding by ~20 fold (**Fig. 6d**), consistent with the results from competition assays (**Fig. 6e-f**). Furthermore, we now detect increased association between UBA6 Δ Cys-Cap and UBE2Z (**Extended Figure 17d**) leading us to hypothesise that the Cys-Cap reduces incoming E2 interactions until it is dynamically altered, potentially caused by ubiquitin loading on the UBA6 active site cysteine. This could explain previous observations that double-ubiquitin loaded UBA1 binds more effectively to E2s (Hershko et al. 1983) . While we believe testing this is outside the current scope of the manuscript, we speculate this point in the Discussion.

- p. 8, line 358: “unique” among which processes? Perhaps “unique among PTMs”?

Thank you for spotting the lack of clarity, this is now corrected.

- p. 8, lines 360: the authors could try avoiding saying “UBA6-activated/specific ubiquitylation” twice in one sentence

Thank you this has been corrected.

- p. 8, line 382 – maybe the authors could define what they mean by extension – extension with respect to what reference UBC domain and how long would it have to be to count as an extension? Should one distinguish between E2s having extra structured domains and just having extra short flexible tails etc.?

Thank you for this point. We have revisited all the E2s and classified them based on extensions exceeding 15 amino acids at the N- and C-termini, as previously described (Wijk & Timmers 2010). We have expanded the table in (**Extended Figure 9b**) to show the number of additional amino acids on either side of the catalytic UBC domain.

- a thought for the discussion: can there be any other reasons why BIRC6 has developed high affinity for UBA6, other than competition with other E2s? Moreover, the authors speak about a stable complex, as judged by SEC, so that might suggest not just fast association but also slow dissociation - can one speculate about some special reasons why would that be? I was intrigued by the cited observation that only a part of UBA6 is Ub-loaded at any given time in the cell. Could UBA6 in its non-loaded state bind to BIRC6 and the two wait together for loading and transthiolation – in this case, would the BIRC6-bound state of UBA6 be equally, or more, or less efficient at binding Ub and ATP and activating Ub? Lastly, how is all this timed with substrate modification - what is the kinetic bottleneck in this cascade? - and could substrate binding further help detach Ub~BIRC6 from UBA6?

Very good points.

We believe the priority mechanism for BIRC6 is required owing to the low abundance of BIRC6 in cells which we mention in the Discussion. The reviewer is correct in enquiring about other reasons – we feel these will be elucidated by further studies into the functional roles of BIRC6 and UBA6.

Building on our analytical SEC results, we have now determined Kds for BIRC6 and UBA6 variants by fluorescence polarisation. These measurements have strengthened our model of specific BIRC6 and UBA6 features controlling recruitment and release.

We believe non-loaded UBA6 does not form a constitutive complex with a significant pool of BIRC6 in cells. Our *in cellulo* pull-downs support this because we only detect a small amount of endogenous BIRC6 with WT UBA6 but this increases with catalytically inactive UBA6, suggesting when BIRC6 binds to UBA6 it receives ubiquitin and dissociates. We now include this in the manuscript.

Regarding kinetic bottlenecks, the cellular concentrations of ubiquitin and ATP are so high that UBA1 and most E2s are constitutively loaded with ubiquitin. The bottleneck therefore would be transfer to the E3 and then to substrates.

- this is probably silly and not worth discussing but could pharmacological strategies be envisioned to trap BIRC6 on UBA6 to target UBA6-dependent cancer? I feel like wherever in biology we have a system that must be released to continue, there is this theoretical potential for getting it trapped, like with PARP1.

Certain cancer cell lines do depend on UBA6 and BIRC6 for survival (Cervia et al. 2022) and thus there is a rationale for targeting them. However, trapping UBA6 and BIRC6 would have significant additional effects on the cell, as all other UBA6-competent E2 pathways would be blocked with unknown consequences.

- p. 51, line 1117: the authors mention that the noncovalent complex actually comes from a UbDha-trapping approach but with a probe with reduced activity. It is good of authors to provide this detail (perhaps it could be explained slightly more), and I wonder if this should be signalled briefly in the main text and figure legend for complete transparency.

We thank the reviewer for appreciating our transparency and we hope they can appreciate it was difficult to mention this in the main text before we reach the section utilising the UbDha probe. We now mention this in the figure legend of **Fig. 2b**.

Further suggestions

I don't think it is a must, and I'm sure the authors have considered it already, but their model would benefit from having a stable mimetic of ubiquitylated BIRC6 or BIRC6 fragment. This would allow directly investigating the binding of such a mimetic – provided it mimics well – to UBA6 in experiments similar to that in Fig. 2a, as well as investigating the competition between ubiquitylated BIRC6 and other E2s in experiments similar to those in Fig. 2g and Fig. 5.

This is an important suggestion from the reviewer, which we are grateful for. We attempted the classical method of producing a stable E2–Ub conjugate first employed by the lab of Ron Hay (Plechanovová et al. 2012) ; however, BIRC6 C4666K was unable to react with Ub. Therefore, we used our UbDha strategy to generate a stable mimetic of BIRC6–Ub by collapsing the E1~Ub thioester with excess DTT. We measured binding of BIRC6–Ub to UBA6 and recorded it is weaker than BIRC6. Owing to the difficulty in making significant quantities of BIRC6–Ub we were unable to use this in competition assays with other E2s. However, our FP data with this mimetic support our thioester switch mechanism.

Secondly, I wonder if it might be possible to investigate the kinetic parameters (k_{on} , k_{off}) of BIRC6 – and potentially also ubiquitylated BIRC6 – binding to UBA6.

We feel a full kinetic understanding of the BIRC6 UBA6 reaction cycle is beyond the scope of this manuscript. However, we have followed this reviewer's suggestions of determining K_d between UBA6 variants and BIRC6 variants / other E2s and making and testing a stable BIRC6–Ub conjugate which all support our mechanistic insights.

In cells, some validation of the model could potentially be gained through co-immunoprecipitation experiments, demonstrating competition between E2s or lack thereof depending on the WT/mutant variants of UBA6 and BIRC6 used. A more final proof would surely involve doing some functional cellular assay to see, for example, if BIRC6 that has lower affinity for UBA6 is not functional.

This is a pertinent question raised by the reviewer. We have performed pull-downs of exogenously expressed UBA6 variants and probed for endogenous BIRC6. As detailed at the beginning of this rebuttal, our results demonstrate that a stable UBA6-BIRC6 complex exists in cells but BIRC6 is constantly being expelled by the thioester switch mechanism. We did consider suitable downstream indicators of BIRC6 functionality, as we agree with the reviewer that this would be the ultimate proof of such an interesting BIRC6-UBA6 interaction. However, the number of validated ubiquitinated substrates of BIRC6 in cellulo is currently limited. We tried to reproduce the UBA6/BIRC6-dependent ubiquitination of LC3B in cellulo, as previously reported (Jia & Bonifacino 2019) . However, in our hands, we could not replicate the changes in LC3B ubiquitination upon depletion of BIRC6 or UBA6, nor even after chemical treatment resulting in degron-induced degradation of BIRC6. We believe more work is required to identify the ubiquitination substrates of BIRC6, and the mutants we have carefully characterised in this study will be a powerful tool for examining the functional roles of BIRC6 and UBA6 in greater detail.

Having listed these suggestions, I don't think these experiments are required; these are just suggestions that came to my mind for how the model could be further tested and expanded, although the authors have thought much longer about it than me and I'm sure they have better ideas for further validating and developing this work.

Reviewer #2 (Remarks to the Author):

Review for NSMB:

This is a very interesting and comprehensive biochemical and structural study investigating how UBA6, which is a second E1 for ubiquitin, and much less well understood than UBA1, functions with E2s. Their careful analyses lead them to propose a hierarchy of E2 preference by E1s, which in my opinion fits well with and extends early work in the field which established how E1's ubiquitin-like fold

functions with specific ubls and E2s. The finding that different E2s need different contributions of E1 architecture is useful.

The data presented are high quality and convincing. The structures are relatively limited in resolution with heterogeneity in ubiquitin conformations in the quaternary complex structures, but the aspects focussed on in the biochemical validation are reasonable and justified.

They show that 2 E2s (ZK1 and BIRC6) are specific for UBA6, and use different mechanisms to achieve the specificity.

In my opinion this study answers an important question as to how the two E1s for ubiquitin work with multiple E2s, from a molecular perspective. As well as this general insight, the main novel finding is the thioester switch mechanism for BIRC6~Ub release – this is an important contribution to understanding E1-E2 interactions and unique adaptations of UBA6 to function with BIRC6.

We thank the reviewer for their time reviewing our manuscript and we are grateful for their support of our study.

I think it could be argued that the evolutionary analysis is somewhat limited, focussing on Drosophila and human – what about other invertebrates? There is also no functional assay in a biological context to explore how the hierarchy they propose works. However, I would consider that outside the scope of this study, which is already a significant contribution.

Our global sequence analysis searched for all species containing UBA6 and BIRC6, whether together or separately, in metazoans. We focused on Drosophila BIRC6 (dBIRC6) due to its characterised functional roles in cell death and development (Domingues & Ryoo 2011; Vernooy et al. 2002). Given the absence of UBA6 in Drosophila, we reasoned there was a high likelihood that dBIRC6 functions with dUBA1, providing a powerful system to tease apart BIRC6-E1 specificity mechanisms. Investigating other invertebrates would be interesting but falls outside the scope of the current manuscript. As a side note, we did attempt to identify residues within BIRC6 required for UBA1 activity by making comparative alignments of BIRC6 orthologues from a range of species with and without UBA6, but we were unable to pinpoint specific determinants of this specificity. As demonstrated in **Fig. 4i and Extended Figure 16**, subtle differences between hBIRC6 and dBIRC6 are sufficient to drive functional differences, notably Glu4604 in hBIRC6 and Asp4663 in dBIRC6, which could not be predicted by sequence analysis alone. We find this striking and highlights the importance of our experimentally derived structures.

Minor points

I'm not sure 'refined to' is the correct term in cryoEM, it is for crystallography, but there is no corresponding 'refinement' in cryoEM.

We thank the reviewer for pointing this out. We have changed "refined to" at the point in the main text where it was leading to resolution. We have retained use of "refined" when referring to data processing such as Non-uniform refinement (Cryosparc) and 3D auto-refine (RELION).

First paper to report ubiquitin fold domain in an E1, and to show that it was required for E2 binding is PMID: 12646924 from 2003, from the Schulman lab.

We apologise for this omission and we have now included this important reference.

Some of the quoted figures seem a bit too precise for the resolution – 899 angstrom squared buried surface area or 214 A² at lower than 3A resolution might be better expressed to the nearest 10 angstroms than the nearest one.

We agree with the reviewer and we have adjusted this accordingly.

Line 169 – abrogated is a legal term, can the authors just say it reduced or prevented binding? The

writing is a little dense in parts, maybe because the text was single spaced it gives me that impression, but I had to read the structure sections several times to get the information.

We thank the reviewer for pointing this out - we have changed the word abrogated. We have also simplified the writing in parts. All text edits are highlighted in green.

Discussion – I don't really understand what a molecular 'rulebook' is – I think they are trying to convey that they have worked out the molecular basis of the specificity of each E1 for its E2s, and that knowing that allows for protein engineering, perhaps framework is a less ambiguous term (and allows for possible future findings that may add to the current understanding!).

We thank the reviewer for this suggestion – we have adjusted this accordingly.

Papers associated with pdb files should be cited (eg in figure legends) as they are experimentally determined and reported.

We apologise for the omission of references for the different PDB structures used. These have now all been included and appropriately cited.

Extended figure 1 legend doesn't include UBA7

Thank you for spotting this. We have removed the reference to UBA7 in the figure legend.

Why do the PDB validation reports say 'not for manuscript review'? Am I supposed to use them in the review or not?

We apologise for this mistake, we had uploaded the PDB validation reports before PDB acceptance. We now include the reports "For manuscript Review".

Reviewer #3 (Remarks to the Author):

In the present manuscript, Riechmann et al investigate specificity determinants governing ubiquitin E1 and E2 enzyme interplay. Through elegant bioinformatic, molecular, and biochemical approaches the authors systematically define sets of E2 enzymes that can be specifically charged by either of the two ubiquitin activating enzymes, UBA1 or UBA6, and identify the E1 domains underlying this specificity.

Consistent with previous results, they find UBE2Z and BIRC6 to be specifically charged by UBA6. They then focus on characterizing the interaction between UBA6 and BIRC6, an E1-E2 pair with previously identified strong genetic interactions. By performing biochemical, biophysical, and cryo-EM analyses, they demonstrate that UBA6-BIRC6 form an unusually strong complex in the absence of ATP and ubiquitin (as compared to other E1 and E2s) and that complex formation is driven by the BIRC6UBC-UBA6UFD domain interaction. This binding mode provides BIRC6 with priority for UBA6 over recognition of other E2s, as demonstrated by in vitro competition experiments. To delve deeper into understanding specificities underlying the UBA6-BIRC6 interaction, the authors next employ an evolutionary-guided approach. They identify that the *Drosophila melanogaster* genome encodes a BIRC6 orthologue (dBIRC6) but only one E1 orthologue that is more similar to UBA1 (dUBA1). They can show that dBIRC6 is charged by dUBA1 and, intriguingly, human UBA1 is able to charge dBIRC6. They then leverage this cross-species E1 reactivity and determine and cryoEM structure of human and *drosophila* E1-UbDha-BIRC6UBC complexes that capture transthiolation intermediates and help explain how UBA6-BIRC6 specificity is achieved. Finally, through analysis of their cryoEM structures coupled with elegant mutational validation experiments, the authors provide evidence for a thioester-switch mechanism as a molecular rationale for how BIRC6, despite its high affinity to UBA6, can be released after ubiquitin transfer.

This is a well-executed and important study, which identifies a previously unrecognized priority mechanism for UBA6-mediated ubiquitin transfer to BIRC6, providing a biochemical rationale for the known strong UBA6-BIRC6 genetic interaction. In addition, this study reveals novel molecular and

structural details underlying activity of subset of ubiquitin E2s with their cognate E1s, UBA1 and UBA6, offering much needed insights into mechanisms that may allow for E1s to orchestrate E2-specific cellular functions. Given the emerging role of E1 enzyme mutations in several human diseases (i.e., spinal muscular atrophy, lung cancer in never smokers, and VEXAS), the results of this study will be of broad interest not only to researchers in the ubiquitin field, but also to those studying protein homeostasis, cell signaling, and disease mechanisms more generally.

We thank the reviewer for their positive assessment of our manuscript.

Below, I am providing questions and suggestion to improve this already very strong study:

1) Molecular rulebook for E1-E2 specificity:

- carefully examining extended data figure 2, I am wondering about the criteria that were used for classifying UBA1-specific E2s in UFD- and UFD- /SCCH-dependent categories, respectively. I.e., were there specific cutoffs used for residual activity with E1 chimera? By eye, it looks like UBE2C still has activity with chimera ii and iv, which is consistent with it being UFD-dependent only. Could the authors please clarify?

We would like to thank the reviewer for bringing this to our attention. We have gone back and carefully re-examined the chimera panels with the E2s and added extra categories to reflect some of the additional differences we observe. Notably, for UBE2C we now classify this outside of UBA1-UFD-specific as we do see some activity with UBA6-1SCCH, which suggests a small contribution from the SCCH domain. We think it will be interesting in the future to examine UBA1-specific E2s utilising both UFD and SCCH domains, and examine the dual-specific E2s that have distinct mechanisms of specificity.

- I understand that for dual specific E2s subclassifying into UFD- and UFD/SCCH-dependent categories is not easily feasible. However, looking over the charging pattern of these E2s with the different chimeras, it looks like that for some E2s (e.g., UBE2L3), UFD and SCCH requirements differ depending on UBA1 and UBA6 context, suggesting that for the same E2, UBA6 and UBA1 may have evolved different recognition strategies. This may be worth discussing in the text.

We agree with the reviewer that some dual-specific E2s like UBE2L3 have very distinct mechanisms governing specificity, and as suggested, we mentioned this in the discussion. It will be interesting in future studies to see how these E2s utilise the different features of UBA1 and UBA6, and we believe our different E1 chimeras provide a good starting point for investigating this.

2) Full length BIRC6:

- As mentioned by the authors, BIRC6 is a ~530kDa protein with many N-terminal domains before the UBC domain. Apart from the initial ubiquitin charging experiments determining that BIRC6 is a UBA6-specific E2 enzyme relying on both UFD and SCCH domain of UBA6 (Extended data figure 2c), all experiments have been performed with the BIRC6UBC domain only. While this is reasonable given the challenges working with such a big protein biochemically, I think it would greatly improve the manuscript if at least some of the competition experiments that address the priority ubiquitin charging mechanism would be repeated with full length BIRC6.

We wholeheartedly agree with the reviewer regarding whether full-length BIRC6 can outcompete the UBA6-selective E2s. We have repeated the priority assay with UBE2D2 and BIRC6 and observed competition of UBE2D2 loading in the presence of full-length BIRC6. This is now included as an additional supplementary figure (**Extended Figure 8b**).

- since a full-length BIRC6 structures (albeit with limited UFD resolution) are available, could the authors conduct molecular modeling/docking studies or at least discuss whether their proposed mechanism for ubiquitin charging of BIRC6 by UBA6 is compatible with the full length BIRC6? In addition, would such studies indicate potential contributions of N-terminal domains of BIRC6 to UBA6-specific charging?

Building on the point raised by the reviewer, we questioned whether any of the additional domains within full-length BIRC6 could contribute towards UBA6 binding. We prepared and screened a cryo-EM sample containing UBA6 and full-length BIRC6 (16 and 8 μM respectively, 1609 micrographs) without addition of the UbDha probe (conditions which should reproduce the non-covalent UBA6-BIRC6 states in **Fig. 2 b-d**). Due to the differences in particle sizes, UBA6 and BIRC6 particles could be picked and processed independently (**Rebuttal Fig. 1 a**). UBA6 particles could be classified into Apo and BIRC6-bound classes, and a low-resolution reconstruction of the BIRC6-bound class corresponded well with the UBA6-BIRC6^{UBC-OUT} model (**Rebuttal Fig. 1 b-c**). Besides the UBC domain, no additional density from BIRC6 was observed in the UBA6-bound class. Conversely, while we could reconstruct the core of full-length BIRC6, neither UBA6 nor UBC domain could be resolved (consistent with all other full-length BIRC6 structures) (**Rebuttal Fig. 1 d-e**). Together, these data are consistent with the UBA6-BIRC6 interaction being confined to the UBC domain of full-length BIRC6.

Rebuttal Fig. 1. Cryo-EM data for full-length BIRC6 with UBA6.

- a.** Manual annotation of BIRC6 and UBA6 particles on a representative micrograph.
- b.** 2D and 3D classification steps used to separate UBA6 particles into UBA6^{APO} and UBA6-BIRC6 classes.
- c.** Front and side views of the UBA6-BIRC6^{UBC-OUT} model fit into the UBA6-BIRC6 volume.

- d. 2D and 3D classification steps used to generate the BIRC6^{FL} consensus volume. Further classification steps did not recover any density for the C-terminal UBC domain.
- e. Front and side views of a BIRC6 model (PDB 8ATX) fit into the BIRC6^{FL} consensus volume (low-pass filtered). Expected location of the flexibly tethered C-terminal UBC domain is indicated with dashed lines. Unknown central density is annotated with an asterisk.

3) Priority mechanism and thioester switch mechanism:

- While the data presented for the UBA6-BIRC6 priority mechanism is convincing *in vitro*, some additional evidence from cells would further strengthen this study. E.g. are there previously published immunoprecipitations experiments showing that BIRC6 is the most abundant E2 enzyme interacting with UBA6 (despite competing with all other UBA6-dependent E2s), or could the authors perform such experiments? Along these lines, compared to WT, would IP of the Cys-Cap UBA6 mutant result in stronger BIRC6 interaction from cells?

We thank the reviewer for suggesting this. Co-dependency analysis in DepMap reveals BIRC6 is the highest co-dependent E2 for UBA6 (Pearson correlation 0.7) with the next highest E2, UBE2Z, showing a much-reduced co-dependency score (Pearson correlation 0.2) (**Fig. 1e**). This indicates a strong functional association between UBA6 and BIRC6 in cellulo. We have now performed pull-downs of exogenously expressed UBA6 variants in HEK293T cells and probed for endogenous BIRC6 as detailed at the beginning of this rebuttal. Consistent with our biochemical and biophysical data, our results demonstrate that a stable UBA6-BIRC6 complex exists in cells; when we prevent thioester release (by using catalytically inactive UBA6) the UBA6-BIRC6 complex is enhanced (**Fig. 6g**). Comparing BIRC6 pull-down with UBA6 Δ Cys-Cap loop showed a marked enrichment, thus our *in vitro*-determined higher affinity to UBA6 lacking the Cys-Cap loop results in a stronger interaction in cellulo. Probing for other E2s, such as UBE2Z, resulted in no discernible stable complex between UBE2Z and full-length UBA6 (catalytically active or inactive) (**Rebuttal Fig. 2**), supporting both our fluorescence polarisation (FP) data of no measurable binding between UBA6 and UBE2Z (**Fig. 2g**) and our finding of BIRC6 priority over other E2s (**Fig. 2h**). Very weak pull-down of UBE2Z to UBA6 Δ Cys-Cap loop was detected (**Rebuttal Fig. 2**), in line with our FP data in which weak binding was measured (**Extended Figure 17d**). This supports our model that the Cys-Cap loop is a general feature that dampens E2 affinity but is crucial to prevent tight-binding E2s, i.e. BIRC6, from becoming a UBA6 inhibitor.

Rebuttal Fig. 2. In agreement with our BIRC6 priority mechanism and FP binding assays, UBE2Z does not form a stable complex with UBA6 WT in cellulo; an interaction is detected with the UBA6 Δ Cys-Cap loop variant but this is still weak in relation to the total pool of UBE2Z in the whole cell lysate (Input samples represent 0.5 % of total cell lysate).

- Could the authors further strengthen their thioester-switch mechanism by introducing mutations in the blocking loop in BIRC6 and providing biochemical evidence for its proposed role in reducing the affinity of the BIRC6~Ub-UBA6 interaction?

We thank the reviewer for this suggestion. Comparison of our structures of BIRC6 binding UBA6 without ubiquitin or receiving ubiquitin (BIRC6^{OUT} vs UBA6~UbDha~BIRC6) identified two pronounced structural features that change: the BIRC6 blocking loop and the UBA6 Cys-Cap loop. These features are located near to each other when no ubiquitin is present (BIRC6^{OUT}). This led us to propose these two loops function together in release of BIRC6~Ub. Based on our FP data and assays testing ability of BIRC6 to out-compete UBE2D2, we have now differentiated two independent roles of these structural features.

We identified a role of thioester release in BIRC6's ability to outcompete UBE2D2 by comparing the competition between BIRC6 WT and C4666A (now presented more clearly in **Fig. 5a,b**). As the reviewer suggested, we designed mutations that alter the BIRC6 blocking loop and assessed their ability to outcompete UBE2D2 and bind UBA6, as described at the start of the rebuttal. The BIRC6 Δ blocking loop, including loss of Trp4673, slightly outperformed wild-type BIRC6 in outcompeting UBE2D2 and even exceeded BIRC6 C466A (**Fig. 5g**), indicating the blocking loop's involvement in thioester release mechanisms. Importantly, this occurred even though the overall affinity between BIRC6 Δ blocking loop and UBA6 was reduced two-fold compared to wild-type BIRC6 (**Fig. 5i**). Focusing on the highly conserved Trp4673, which blocks access to BIRC6's active site Cys before ubiquitin transfer, W4673A marginally increased competition relative to C4666A and bound slightly more tightly, while W4673R (which we would predict to alter the conformation of the blocking loop and/or the surrounding region) showed reduced competition compared to WT and C4666A BIRC6. However, we could not detect any binding of W4673R to UBA6 in FP assays and yet this mutant displayed more competition of UBE2D2 than the UFD null-binding mutant A4575D. This, therefore, additionally supports a role for the blocking loop, and furthermore its conformation, in thioester release. Overall, these findings demonstrate that the blocking loop has multiple effects on the UBA6 transthiolation reaction. On one hand, the blocking loop contributes to binding UBA6, and its deletion reduces this interaction, or altering its conformation can prevent binding altogether in our assays. Additionally, losing/altering the blocking loop impairs the thioester switch mechanism. In the results and discussion, we describe these different roles of the blocking loop but caution that other features within BIRC6 or the ubiquitin itself in the BIRC6~Ub complex may contribute to thioester release.

- As mentioned by the authors, a recent study (PMID: 39143218) has shown that E1-to-E2 transthiolation is driven by coupling adenylation and transthiolation. Could the authors discuss in a bit more detail how this would integrate with their model of the thioester switch mechanism for UBA6?

The study PMID: 39143218 used an E2 with no extended blocking loop (Ubc4, yeast UBE2D2). Therefore, the blocking loop mechanism for BIRC6 is likely not required for the coupling of transthiolation to adenylation in this instance, and as mentioned above, our discussion highlights the different thioester release mechanisms employed by other E1-E2 pairs.

- Out of curiosity, since the E1-E2 and E2-E3 surfaces partially overlap, would the various E2 mutants that modulate E1 specificity generated by this study also interfere with E3 interactions?

We thank the reviewer for highlighting this important point. We have examined several E2-E3 structures across different E3 families, all using the common dual-E1 functioning E2s UBE2D2 and UBE2L3. Although there is overlap in E1 and E3 interaction sites on these E2s, the residues we mutated in UBA1-specific E2s (UBE2E1, UBE2H, and UBE2W) to promote UBA6 specificity, as well as those in UBE2D2 to dampen UBA6 dual activity, are located on the opposite side of helix 1 and do not interact with the respective E3s. This finding emphasises the effectiveness of our mutational approach in changing E1 specificity and provides a useful tool for studying specific E1-E2 cellular pathways. We have added a new supplementary figure (**Extended Figure 10a-h**) illustrating the positions of the residues we mutated in UBE2D2 and their location along helix 1 relative to residues involved in E3 interactions.

Statistical analysis:

- Statistical analysis should be included for Fig 1c, Fig 3c,e,f, Fig 5c,e

Following the reviewer's suggestion, we have now included statistical analysis for Fig. 3c,e,f, Fig. 5c,e, and Extended Figure 15e,f (formerly Extended Figure 14e,f). For the analysis in Figure 3, we aimed to compare the differences in specificity switch between UBA1 and UBA6. Therefore, we now include additional differences in the loading of each E2~Ub and perform the appropriate statistical test on these. For Fig. 1c, we wanted to highlight the relative differences in E2 selectivity for UBA1 and UBA6 rather than compare the differences between each E2, as the relative activities of each E2 vary.

Figure reference and legend errors and suggestions to improve figures:

- Fig 3a: please explain the color coding of the residues in the legend.

By default, the Clustal colouring scheme omits positions below a certain identity threshold. We agree with the reviewer that this is not helpful for our figure and therefore have changed the colouring scheme in all sequence alignments so that every position is coloured by residue property.

- Fig 3b: the words “additional FITC-labelled gel image” in the figure legend should go with 3c

Thank you this has been corrected.

- Fig 3b: consider highlighting the steric clashes and favorable interactions with some dotted lines or distance information

We appreciate this reviewer's suggestion, but we found it difficult to convey favourable and non-favourable contacts, especially since most of the non-favourable contacts are not due to steric clashes but charge differences. Additionally, we do not wish to display distance information, as these are predictive models that lack experimentally derived parameters.

- In section “BIRC6-UBC-UBA6-UFD interaction explain specificity for UBA1-specific E2s” extended figure 3c should refer to 2c

Thank you this has been corrected.

- Extended data figure 1 legend indicates that UBA7 is in the alignment, but UBA7 is not shown. There are some rows with UBA7 labels. Please correct this figure and also explain color coding

We apologise for this mistake and thank the reviewer for spotting this. We had previously included UBA7 in the alignment but removed it for clarity. We have now removed the UBA7 labels and annotated where the alignment colours come from. As mentioned above, all positions are now coloured by residue property.

- Extended data figure 8: the figure legends do not match up with the legends. Please also indicate which experiments are single or multi-turnover reaction. Should be cleared up.

Thank you this has been corrected.

References

Cervia LD, Shibue T, Borah AA, Gaeta B, He L, et al. 2022. A Ubiquitination Cascade Regulating the Integrated Stress Response and Survival in Carcinomas. *Cancer Discov.* 13(3):766–95

Domingues C, Ryoo HD. 2011. Drosophila BRUCE inhibits apoptosis through non-lysine ubiquitination of the IAP-antagonist REAPER. *Cell Death Differ.* 19(3):470–77

Hershko A, Heller H, Elias S, Ciechanover A. 1983. Components of ubiquitin-protein ligase system. Resolution, affinity purification, and role in protein breakdown. *J. Biol. Chem.* 258(13):8206–14

- Jia R, Bonifacino JS. 2019. Negative regulation of autophagy by UBA6-BIRC6-mediated ubiquitination of LC3. *Elife*. 8:e50034
- Plechanovová A, Jaffray EG, Tatham MH, Naismith JH, Hay RT. 2012. Structure of a RING E3 ligase and ubiquitin-loaded E2 primed for catalysis. *Nature*. 489(7414):115–20
- Vernooy SY, Chow V, Su J, Verbrugghe K, Yang J, et al. 2002. Drosophila Bruce Can Potently Suppress Rpr- and Grim-Dependent but Not Hid-Dependent Cell Death. *Curr Biol*. 12(13):1164–68
- Wijk SJL van, Timmers HTM. 2010. The family of ubiquitin-conjugating enzymes (E2s): deciding between life and death of proteins. *FASEB J*. 24(4):981–93

Our ref: NSMB-A50827A

7th Oct 2025

Dear Dr. Elliott,

Thank you for submitting your revised manuscript "Molecular basis of UBA6 specificity for ubiquitin E2 conjugating enzymes reveals a priority mechanism of BIRC6" (NSMB-A50827A). It has now been seen by the original referees and their comments are below. The reviewers find that the paper has improved in revision, and therefore we can now accept it in principle in Nature Structural & Molecular Biology, pending revisions to satisfy the referees' final requests and to comply with our editorial and formatting guidelines.

To facilitate our work at this stage, it is important that we have a copy of the main text without the figures as a word file. If you could please send along a word version of this file as soon as possible, we would greatly appreciate it; please make sure to copy the NSMB account (cc'ed above).

Sincerely,

Dimitris Typas
Senior Editor
Nature Structural & Molecular Biology
ORCID: 0000-0002-8737-1319

Reviewer #1 (Remarks to the Author):

I thank the authors for their thoughtful responses and for revising their manuscript. The new experiments, including those involving the thioester mimetic and the blocking loop mutants - and the two new approaches (fluorescence anisotropy and cellular pulldown) - complement and strengthen the manuscript.

We thank the reviewer for their positive evaluation of our revised manuscript including our additional experimental approaches.

Below, I list a few last minor things I noticed; I think the manuscript is otherwise ready for acceptance.

verse 147: "which" missing before "could"?

We agree and have now altered the text.

verse 203: "analyses offers" -> "analyses offer"

Thank you, we have now altered the text.

verse 338: Could the authors illustrate how the mimetic was made using UbDha (a similar scheme to that in Ext Dat Fig 11d)? Perhaps it is provided somewhere but I missed it.

We agree that a diagram illustrating this would be useful to include to complement the existing description in the Methods section "Fluorescence Polarisation". This is now in Supplementary Figure 5a.

verse 339: The 1.5-fold difference in binding between the Ub-BIRC6 mimetic and BIRC6 alone is

modest, especially given the relatively large errors in the estimated Kd values provided in Fig. 5c. I think it is still consistent with the authors' model, but they could comment on the remaining uncertainty. It might be worth giving the estimated Kd values together with errors in the main text and not only in the figure.

We now state the Kd values with the error range in the text.

verse 403: Consider adding the part between asterisks for better clarity: "However, this increase in binding is still not sufficient to enable priority *of UBE2Z or BIRC6 A4575D* over UBE2D2 (Fig. 6e,f) compared to BIRC6 WT".

We agree and have now altered the text to clarify this.

verse 518: The sentence starting with the following phrase is a bit convoluted - consider revising and perhaps dividing into two sentences: "We therefore propose that the Cys-Cap loop may...".

We agree and have now altered the text to simplify this.

verse 966: This is not important, but it might be interesting to do ConSurf colouring based on MSA of BIRC6 orthologues only from species that have UBA6.

We agree that sequence variations in the blocking loop could track with the presence of UBA6. However, comparing blocking loops in Human BIRC6 vs Drosophila BIRC6 (there is no UBA6 in Drosophila), the sequences are identical (NTGHGR). This might reflect multipurpose roles of the blocking loop, similar to what we describe for other structural features within E1s and E2s.

verse 1281: "no Kd could be fit" -> "no Kd could be fitted"

We agree and have now altered the text.

verse 1508: there is a rectangle symbol after MgCl (instead of 2 in subscript?)

We thank the reviewer for so astutely spotting this. We have corrected this formatting issue.

Suggestion for the future

I believe this goes beyond the scope of the current manuscript, but I wonder if the fluorescence anisotropy assay could be used in the future to test displacement of fluorescently-labelled BIRC6 or Ub-BIRC6 by their unlabelled counterparts provided in excess to estimate koff values for the labelled compounds. One could also test displacement of Ub-BIRC6 by BIRC6. Furthermore, I wonder if it might be worth repeating the measurement of Kds for Ub-BIRC6:UBA6 and BIRC6:UBA6 in the presence of an excess of free Ub that might help outcompete Ub-BIRC6 in a physiologically-relevant manner.

This is an excellent suggestion and we will certainly consider it for our future studies.

Reviewer #2 (Remarks to the Author):

The authors have satisfactorily addressed all my previous comments and concerns.

We thank the reviewer for their positive response to our revised manuscript.

Reviewer #3 (Remarks to the Author):

In the revised version of their manuscript, Riechmann et al. have done an excellent job in carefully and comprehensively addressing all of the points I raised in my original review. In particular, they clarified classification criteria and expanded on mechanistic distinctions for dual-specific E2s, repeated key competition assays with full-length BIRC6, and strengthened their proposed priority and thioester-switch mechanisms with new biochemical and cellular evidence. This is an important study

that not only provides new mechanistic insight but also generates valuable tools to study E1–E2 interactions in the future. I congratulate the authors on their impactful work.

We thank the reviewer for their positive assessment of our work.